# Does increasing biodiversity in an urban woodland setting promote positive emotional responses in humans? A stress recovery experiment using 360-degree videos of an urban woodland

**Simone Farris**[1]*, **Nicola Dempsey**[1], **Kirsten McEwan**[2], **Helen Hoyle**[1], **Ross Cameron**[1]

**1** Department of Landscape Architecture, The University of Sheffield, Sheffield, United Kingdom, **2** College of Health, Psychology and Social Care, University of Derby, Derby, United Kingdom

* sfarris1@sheffield.ac.uk

## Abstract

Green spaces can support human stress reduction and foster positive emotional well-being. Previous research has suggested that biodiversity (i.e. the variety of species of plants and animals in a given location) can enhance recovery from stress even further. However, there is limited experimental evidence testing this hypothesis and results, to date, have been mixed. This study aimed to provide further understanding of the role of biodiversity (actual or perceived) on human well-being by experimentally manipulating species richness and stress. Participants (372 in total) took part in an online experiment, where they received an episode of mild stress before watching a 360-degree video to recover. The video showed the same location, an urban woodland, but at one of four artificially manipulated levels of biodiversity. The participants reported their Positive and Negative Affect before and after the stress induction and after watching the video, providing a measure of their stress and well-being throughout the experiment. Participants also reported their perceptions of biodiversity (i.e. how diverse they thought the location was) and elaborated on their responses with brief comments. Repeated Measure Analysis of Variance revealed that exposure to all levels of biodiversity reduced the participants' Negative Affect, but with no significant difference between the conditions. However, the analysis showed higher Positive Affect in those participants who perceived the environment as more biodiverse. Comments from participants indicated that those who reported noticing flowers and trees in the environment also showed higher Positive Affect. This suggests that perceiving biodiversity promotes more positive emotions, but critically one needs to actually notice (engage with) the components of biodiversity to elicit these extra benefits.

## Introduction

Urban green space has been cited as providing some degree of protection against poor mental health [1–3] and providing recovery opportunities for individuals suffering from mental health

**Data Availability Statement:** The dataset file is available from the UK Data Service database URL:

https://reshare.ukdataservice.ac.uk/856311/ DOI: https://dx.doi.org/10.5255/UKDA-SN-856311.

**Funding:** SF received a PhD Studentship funding from the UKRI Economic and Social Research Council, via the White Rose Doctoral Training Partnership (WRDTP). Award reference: 2273610. https://www.ukri.org/councils/esrc/ The funders had no role in study design, data collection and analysis, decision to publish, or preparation of the manuscript.

**Competing interests:** The authors have declared that no competing interests exist.

problems [4–7]. Several theories have proposed explanatory mechanisms behind these benefits, which rely on the content or on the quality of green space. The Attention Restoration theory [8] postulates that space must provide engaging but low-effort stimuli (soft fascination) to be restorative. The Stress Reduction theory [9] suggests that spaces with unthreatening nature and moderate complexity reduce stress most effectively. The Biophilia hypothesis [10] indicates human well-being is enhanced by interacting with other species. More recently, research work addressed some of the physical aspects associated with green space such as phytoncides [11] and beneficial microbial communities which affect the human immune system and mood regulation [12, 13].

It has been argued that biodiversity (i.e. the diversity of species of plants and animals in a given location) is an attribute of the landscape [14] and one of the distinctive qualities of urban greenspaces [15]. Biodiversity can be quantified with objective measurements (i.e. actual biodiversity) but also estimated subjectively (perceived biodiversity), based on the number of species an observer thinks could be in the location [16]. According to the theories referenced above, a green space that is (or is *perceived* to be) richer in species should provide more opportunities to encounter wildlife and promote soft fascination (Attention restoration), and interactions with unthreatening nature (Stress reduction).

Indeed, a number of papers suggest that increased bird [15, 17–19], invertebrate [15] and/or plant diversity [15, 17, 18, 20] increases the health-promoting potential of the landscape. Biodiversity in green space has been linked to both reduced stress [21–23] and enhanced Positive Affect [18, 19].

However, Lovell et al. [24] and Marselle et al. [25] state that the links between biodiversity and health are often correlational, and recommend more experiments to better explore cause and effect. Botzat et al. [26] also argued that, in urban green spaces, biodiversity was more frequently studied at the ecosystem and habitat level than at the species level. A recent review [27] noted that only six studies (out of the 52 reviewed) were set up to specifically explore the influence of species richness on human health. Two studies were natural experiments, using locations at different levels of biodiversity. Hussain et al. [28], measured self-reported well-being, blood pressure and heart rate before and after exposing participants to six mountain meadows. Participants reported higher well-being after visiting meadows with high plant biodiversity, while blood pressure and heart rate did not differ. Simkin et al. [29] tested the effect of visiting forests on Attention Restoration, Vitality and Positive/Negative Affect. The participants visited four forests which were all dominated by spruces (*Picea abies*), but differed in terms of maturity (old vs young) and location (urban vs rural). The results showed that rural mature forests (i.e. more biodiverse) induced higher Attention Restoration, Vitality and Positive Affect than young or urban forests. Four experiments manipulated species richness directly. Wolf et al. [18] showed videos comparing tree (1 vs 4 species) and bird (1 vs 5 species) species richness and found that videos with more species reduced anxiety and increased Positive Affect and Vitality. In contrast with Hussain et al. [28], the creation of *urban* meadows at three levels of species richness (low, medium and high) elicited non-significant differences in self-reported physical health and mental well-being, compared with mown grass [30].

The remaining two experiments linked increased biodiversity levels to better stress recovery. Placing stressed individuals in front of one out of five arrangements of potted plants (with 0, 1, 16, 32 and 64 plant species respectively), showed optimal stress recovery (regulation of blood pressure) with the 32 species treatment [22]. Schebella et al. [31] also indicated that *some*, but not necessarily *high* biodiversity, reduced stress. They used 360-degree videos to both stress their participants and expose them to parks at different levels of species richness. Applying a multisensory approach, species richness was controlled via visual (2, 4, 7 vegetation

layers), audio (birdsong from more or fewer species) and olfactory stimuli (1 to 3 smells from grasses species). Results showed that the low biodiversity scenario (2 vegetation layers, 1 bird, 1 smell) lowered anxiety and heart rate, compared against an urban control (i.e. little biodiversity) but also the treatments representing greater biodiversity.

Another strand of studies has found that improved psychological well-being was more associated with people's *perceptions* of biodiversity, rather than with the actual species richness [32]. Using measures of both the actual and perceived richness of trees, butterflies and birds in twelve urban parks in Lisbon, Gonçalves et al. [33] found that perceived species richness explained more of the variance of well-being (Attention Restoration) than actual species richness. Similarly, students perceiving higher animal species richness in the parks of Singapore reported higher attention restoration and Positive Affect [23].

Perceptions of biodiversity are experienced through the senses [26] and several factors may influence these perceptions. These include visual clues such as planting height [30], diversity of flower colours [34] and broadleaf shapes [33]; auditory clues such as birdsong and sounds from water [35, 36]; and olfactory clues such as smells of understory plants and fungi [37]. Perceptions of biodiversity, however, have at times been reported to be closely aligned with recorded data sets [30, 31, 38], at other times overestimated [39] or poorly correlated with actual biodiversity [32, 40]. This may be due to difficulties in conceptualising biodiversity outside the expertise of ecologists [41].

Overall, these studies provide evidence of a positive link between species richness (actual or perceived) and some health indicators. Although limited by the number of studies, the experimental evidence suggested this relationship could be non-linear (i.e. a moderate level of diversity is more beneficial than too high or too low). However, since many studies have compared the effects of species richness present in different locations (e.g. different parks), the specific contribution of species richness to health and well-being remains unclear. Although different locations or settings allow for an easier (possibly more realistic) comparison of species richness, it has been argued that as the location changes, so do other characteristics, such as landscape size, heterogeneity, naturalness, management and other factors that influence landscape quality, e.g. the presence of water [19, 31].

## The present study

The study presented here aimed to provide further understanding of the role of biological diversity on human well-being by manipulating species richness and stress while controlling for the effect of the location. Similarly to Lindemann-Matthies and Matthies [22] one single locational setting (a woodland) was manipulated to increase the level of plant species richness by incorporating additional plants. Digital video recordings (360˚) of each scenario (condition) were then taken for participants to view at a later date.

Videos were used because exposure to the natural environment can be simulated through visual and audio media and elicit positive health responses [42–46]. Interactive, 360-degree videos are emerging as a novel method to expose participants to green spaces [47, 48] and allow viewers to rotate the camera in any direction and explore the location. Such videos can mimic *in-situ* interactions, with participants who watched 360-degree videos of monuments [49] or scenic lakes [50] reporting similar spatial presence and emotional reactions to those who actually visited these sites. Understandably, however, direct exposure to the natural environment generates a stronger affective response, compared to any video-mediated exposure [51, 52].

The use of videos to record the location, but with and without additional plants or bird songs, meant that all other landscape factors were consistent across all video treatments, i.e.

the only factors that varied were the ones we artificially manipulated. This allowed us to test the following hypothesis:

- Increased plant diversity corresponds to improved Positive Affect and reduced Negative Affect after a stressful event.
- Increased bird diversity corresponds to improved Positive Affect and reduced Negative Affect after a stress event.

Finally, given that perceptions of biodiversity could be more important than actual species richness in producing well-being, we measured perceived biodiversity to test if:

- Higher perceived biodiversity corresponded to improved Positive Affect and reduced Negative Affect after a stress event.

## Methodology

### Condition set up and video recording

Plant biodiversity was manipulated by creating four different conditions for the one woodland location in Sheffield (UK). The site is managed as an urban conservation zone and is approximately 200 x 150 m in size. Access to the site was granted by the Estates and Facilities Management of the University of Sheffield. A botanical survey in May 2021 found 36 vascular plant species in the woodland, including non-native species (S1 Appendix). A camera was set up on a southeast sloping site in the northwest sector of the woodland, and only recorded natural features (e.g. the presence of vehicles, houses etc. were excluded). The camera lenses were placed at 1.3 meters from the ground, simulating an average eye level for a sitting person.

Condition 1 acted as a control and was filmed without introducing any additional plants. For condition 2 (low plant diversity), 60 individual plants of 4 taxa were introduced to the woodland within view of the camera (Table 1). These plants were then removed and for conditions 3 and 4 (high plant diversity) replaced by 60 other individuals representing 23 taxa (Table 1). Small-flowered (i.e. non-flamboyant) varieties were used to represent typical woodland plant types. The plants were placed alongside existing natural herbaceous vegetation, avoiding excessive shade, pathways or where there might be a strong contrast between the flowers' colours (Fig 1) in an attempt to maintain an illusion of naturalness.

In conditions 1–3, the audio from the footage was replaced with the sound of a gentle breeze (Table 2). This provided a credible background sound for a woodland while controlling for the effects of plant (visual) and bird (auditory) diversity. Condition 4 was produced with the footage of condition 3 and audio recorded in the same location but on a different day. This featured birdsong from 6 species (S1 Appendix) and water sounds from a nearby stream (Table 2). The sound was recorded with a Zoom H1 (Zoom Corporation, Tokyo, Japan), set to record at a sampling frequency of 48kHz at 16 bits. All footage was captured with the camera Insta360 OneX (Arashi Vision Inc., Shenzhen, China), at 5.6k resolution and 30 frames per second and leaving the rest of the settings on auto. Adding sounds to all videos was a necessary step to simulate the woodland on screen. Participants can experience "restlessness" and feel "cut off" from the experience of nature when watching a slideshow of a woodland with no sounds [53].

### Participants' recruitment and ethics statement

Participants were recruited through social networks (Facebook and Twitter), via a University of Sheffield Volunteer mailing list (including staff and students), and on two survey exchange

**Table 1. Plant taxa, main flower colour and number of specimens used in conditions 2, 3 and 4.**

| Species/cultivar | Flower colour | Condition 2 Low plant Diversity (no. of specimens) | Conditions 3 and 4 High plant Diversity (no. of specimens) |
|---|---|---|---|
| *Anchusa capensis* cv. Blue Angel | blue | | 2 |
| *Achillea* cv. Moonshine | yellow | | 2 |
| *Bidens ferulifolia* cv. Blazing Glory | orange | | 3 |
| *Brachyscome multifidi* | pink | | 3 |
| *Diascia* cv. Diamond Fuchsia | pink | 12 | 6 |
| *Lavandula angustifolia* | lilac | | 2 |
| *Lobelia cardinalis* cv. Queen Victoria | red | | 2 |
| *Lysimachia nummularia* | yellow | | 2 |
| *Nemesia* cv. Berries and Cream | purple | | 3 |
| *Nemesia* cv. Framboise | lilac pink | | 3 |
| *Osteospermum* cv. Akila Purple | purple | | 2 |
| *Osteospermum* cv. Akila Yellow Shades | yellow | | 2 |
| *Petunia* cv. Frenzy Yellow | pale yellow | | 3 |
| *Phlox* cv. Dwarf Beauty Blue | blue | 24 | 3 |
| *Phlox* cv. Dwarf Beauty Scarlet | red | | 5 |
| *Phlox* cv. Dwarf Beauty White | white | 18 | 3 |
| *Salvia nemorosa* cv. Caradonna | blue | | 2 |
| *Sanvitalia* cv. Aztec Gold Hussare Knob | yellow | 6 | 3 |
| *Tradescantia* cv. Blue 'n' Gold | blue | | 1 |
| *Verbena* cv. Showboat Dark Red | red | | 2 |
| *Verbena* cv. Showboat Midnight | deep purple | | 2 |
| *Verbena* cv. Showboat Pink | pink | | 2 |
| *Verbena* cv. Showboat White | white | | 2 |

platforms (Surveyswap.com and Survecircle.com). Data collection took place from July 2021 to December 2021. No incentives or rewards were offered for participating. After being informed about the study procedure, all participants provided anonymous written consent by completing the consent form. None of the authors had access to information that could identify individual participants during or after data collection. The study and the procedure were

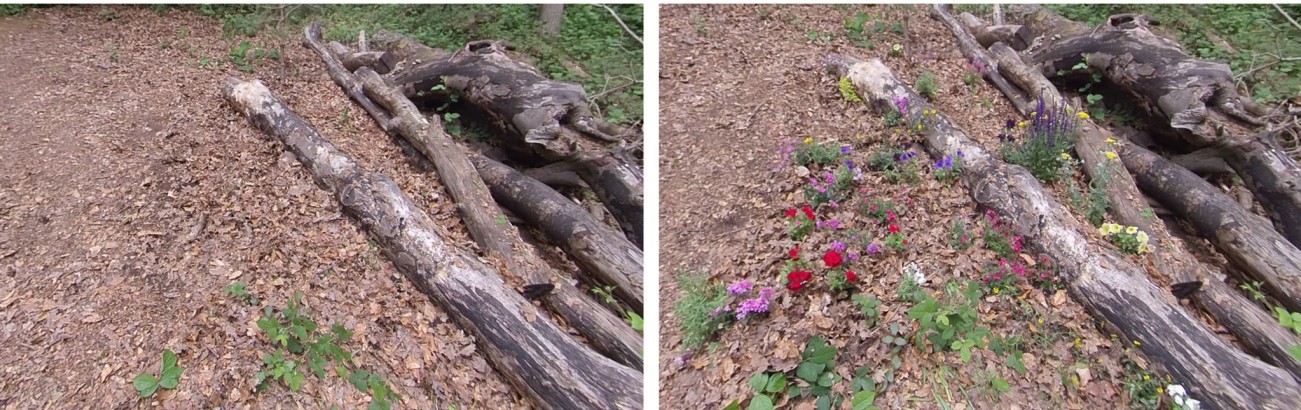

**Fig 1. Photograph of the filming site taken under control conditions (left) and after adding 21 species of plants (right).**

**Table 2. Experimental conditions.**

| Condition | No. of species added | No. of individual plants added | Sounds |
|---|---|---|---|
| 1—Control | 0 | 0 | Gentle breeze |
| 2—Low plant diversity | 4 plants | 60 | Gentle breeze |
| 3—High plant diversity | 21 plants | 60 | Gentle breeze |
| 4—High plant diversity and a diverse natural soundtrack | 21 plants, 8 Bird songs | 60 | Birdsong and water |

reviewed and approved by the Department of Landscape Architecture Ethics Committee (Approval Ref. 039698).

## Experimental procedure

Online participants were asked for demographic information and their affective states (emotions) before and after 2 interventions–the first being a stress induction (a loud noise) and the second a video designed to provide a degree of relaxation and calm (Fig 2). Participants were randomly assigned to one of the 4 videos representing the 4 different conditions outlined above.

In stage 1, the participants provided their demographics and their baseline affective states; the affective questionnaire's items (see Measures, below) were always presented in random order. Stage 2 began immediately after completing the affective questionnaire. Upon loading the next survey section, the sound of a fire alarm (a stressor) was played for 15 seconds. As required by the ethics review, participants were pre-warned that they would be listening to an intrusive, annoying noise. Participants were then asked again to rate their affective states after experiencing the stressor. In Stage 3, participants were randomly assigned one of the four videos to watch. In-video instructions reminded them to activate the full-screen and how to move the camera around. A timer-controlled button prevented the participants from skipping ahead, allowing them to proceed only after 330 seconds had passed. After the video, the participants rated their affective states one last time and were asked for their perceptions and comments.

## Measures

The affective states were measured via the International Short Form of the Positive and Negative Affect Schedule (I-PANAS-SF, Thompson, 2007). This questionnaire has ten items, five measuring Positive Affect states and five Negative Affect states; the items are rated on a 5-point Likert scale, indicating how much the participant is feeling an emotion from "not at all" to "extremely". The two main dependent variables, the Negative Affect (NA) and the Positive Affect (PA), were computed by summing the scores from the I-PANAS-SF scales,

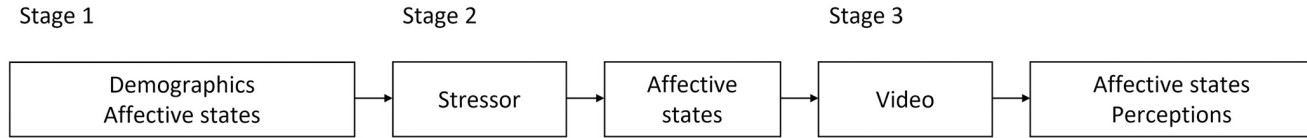

**Fig 2. Experimental procedure with affective states being measured at three discrete points, i.e. before a stressor, after a stressor and after a video of green space.**

producing two semi-continuous variables ranging from 5 to 25. The internal reliability, measured as Cronbach alpha, was good for both scales (Cronbach $\alpha > 0.8$). The I-PANAS-SF was chosen as it has a short completion time, is validated for non-native English speakers and has good reliability for representing genuine emotions [54]. Affective states, which include emotions, stress responses and mood, are important indicators of mental health [55]. Short-term affective responses have been linked to longer-term mental health indicators, such as life satisfaction and depression [56, 57], thus making the I-PANAS-SF a useful proxy for determining potential health outcomes related to nature-based interactions.

Perceptions of biodiversity were measured at the end of stage 3 (i.e. after watching the video). The participants were asked to rate the location shown in the video in terms of "value for plants and wildlife" [58, 59]. The rating ranged from "very bad" (1) to "very good" (5). Those who rated the environment as "good" or "very good" were also asked to share what they noticed in the video that corresponded to this response (open question). This provided a proxy of their perceptions of biodiversity without mentioning the word directly, reducing the risk of biased responses.

Participants were asked to provide their demographic characteristics (Gender, Age, Ethnicity), to verify the balance of the randomisation (i.e. all condition groups should have similar demographics). To check for the novelty effect participants were also asked if they had ever watched a 360-degree video before.

Along with their demographics, participants were asked to rate their Nature Connectedness via the Inclusion of Nature in Self scale (1–7 scale) [60] and to recall the amount of time they spent outdoors during their childhood, ranging from "none" (1) to "a lot" (4). Nature Connectedness, one's extent of affective affiliation with nature [61], has been shown to act as a moderator of the well-being benefits derived from being in nature [62].

## Statistical analysis

All analyses were carried out on SPSS (version 26) for Windows. Metadata (completion time, IP location) were used to check the quality of the data. Responses meeting one or more of the following criteria were excluded: 1) Incomplete response; 2) Overtime completion (over 30 minutes); 3) Invalidating comments (e.g. the participant admitted clicking through the procedure); 4) Multiple submissions (same IP and demographics). Only the first valid observation was kept; 5) All I-PANAS-SF ratings, at all stages, were the same number (e.g. all 1).

One-way analysis of variance (ANOVA) was used to compare the condition groups in terms of the demographics, at baseline. Changes in Negative Affect (NA) and Positive Affect (PA) over the three stages of the experiment and between the condition groups were examined via repeated measure analysis of variance (RM-ANOVA). Since the procedure was identical for all participants except for the video, the RM-ANOVA model included an interaction term between the stage of the procedure and the between-subjects variable. Where the RM-ANOVA assumption of sphericity was violated, probability (*P*) values were calculated with the Huynh-Feldt correction [63]. Where post-hoc comparisons were performed, the Bonferroni correction for multiple comparisons was applied.

Video condition was selected as the main independent variable (between groups). Separately, the perceived biodiversity of the site, the participants' Nature connectedness and time spent outside during childhood were also used as independent variables.

To identify the effect of individual elements of nature on the affective states, comments from participants who had evaluated the environment as "good" or "very good" were analysed via Content Analysis [64], generating a list of codes (S1 Appendix). All codes were used as dummy variables (between groups) in a series of exploratory RM-ANOVA analyses.

To ensure statistical validity, the required sample size was estimated a priori using power analysis on G*Power (version 3.1.9.7; [65]). The estimated sample size was N = 324, considering the analysis design (repeated measures), a significance level of α = 0.05 and 80% power and expecting small effect sizes (Cohen's d = 0.15; [66]). This expectation was informed by previous meta-analyses, which reported moderate to small effect sizes on the affective states [52] and smaller effects when the nature exposure was mediated by video [51] compared to a real nature location.

## Results

### Descriptive data

There were 602 responses to the survey, 414 of which were complete (68% completion rate). After excluding those responses that did not meet the required criteria (see above), 372 responses were analysed.

Females comprised 63% of respondents and the age distribution was skewed towards groups aged 18–24 (43%) and 25–34 (34%). The ethnicity was predominantly White (66%) followed by Asian (19%) and Mixed (8.3%). Black and other ethnic groups collectively represented 6% of the sample. Two-thirds of the participants (69.7%) declared to have watched a 360-degree video before. Most of the participants took the survey on a laptop (64.4%) or a desktop (23.4%), as recommended, while fewer used a phone (10.8%) or a tablet (1.3%).

In terms of engagement with nature and outdoor space, 47.8% of the participants recalled having spent 'a lot of time outdoors' in their childhood while 41.7% recalled having spent 'a moderate amount of time' outdoors.

The mean value for the Inclusion of Nature in Self was 4.05 (SD = 1.64), however, most of the responses fell on scores of '5' (27%) or '2' (19%), suggesting some polarisation around those who considered nature as an important part of their lives, and those who felt more removed from nature.

Across all treatments, the perceived value of the environment for plants and wildlife (hereafter referred to as "perceived biodiversity") was rated "good" by 53% of participants, followed by "very good" (23.4%), "neither good nor bad" (18.5%) and "bad" or "very bad" (5% collectively). To ensure statistical balance, the three lower categories were grouped together in subsequent analysis ("i.e. bad or neutral). On average, the completion time of the procedure was 13 minutes.

### Affective responses to the biodiversity in the videos

All video conditions reduced Negative Affect (NA). Negative affect scores varied significantly with time (i.e. stages; p < 0.005; Table 3), but not due to condition (p = 0.31). For all conditions, NA scores increased significantly after the stressor but decreased significantly after watching (any of) the videos (Fig 3, left). Increasing the amount of biodiversity or natural sounds in the videos had no significant effect on reducing negative scores.

Pooling data across the conditions showed that Positive affect (PA) scores changed with time (p = 0.02). Post hoc tests showed that PA scores were statistically lower after the stressor (p = 0.014) but not after the relaxing video (p = 0.14), compared to the baseline (Fig 3, right). There was no significant difference in the PA scores before and after the condition videos. There were no significant differences between any of the experimental groups at any stage (p = 0.91), i.e. the amount of additional biodiversity or natural sounds were not affecting the Positive Affect scores, compared to a control.

**Table 3. Summary of the RM-ANOVA models showing the effect of several predictors on the affective scores.**

| Dependent variable | Analysis component | F | p-value | Partial eta squared |
|---|---|---|---|---|
| **Effect of the video's biodiversity** | | | | |
| Negative Affect | Within -subjects | 141.6 | < 0.005 | 0.28 |
| | Between Subjects | 1.2 | 0.31 | 0.01 |
| | Interaction | 1.02 | 0.41 | 0.008 |
| Positive Affect | Within-subjects | 3.94 | 0.02 | 0.01 |
| | Between Subjects | 0.18 | 0.91 | 0.001 |
| | Interaction | 1.18 | 0.31 | 0.01 |
| **Effect of the perceived level of biodiversity** | | | | |
| Negative Affect | Within-subjects | 114.7 | < 0.005 | 0.24 |
| | Between Subjects | 3.43 | 0.03 | 0.02 |
| | Interaction | 1.24 | 0.29 | 0.007 |
| Positive Affect | Within-subjects | 3.13 | 0.04 | 0.008 |
| | Between Subjects | 6.93 | < 0.005 | 0.04 |
| | Interaction | 6.74 | < 0.005 | 0.03 |
| **Effect of noticing flowers** | | | | |
| Negative Affect | Within-subjects | 60.45 | < 0.005 | 0.14 |
| | Between Subjects | 0.73 | 0.79 | < 0.005 |
| | Interaction | 1.07 | 0.3 | 0.003 |
| Positive Affect | Within-subjects | 3.36 | 0.04 | 0.009 |
| | Between Subjects | 1.75 | 0.19 | 0.005 |
| | Interaction | 5.35 | 0.005 | 0.01 |
| **Effect of noticing sounds** | | | | |
| Negative Affect | Within-subjects | 75.87 | < 0.005 | 0.17 |
| | Between Subjects | 6.62 | 0.01 | 0.02 |
| | Interaction | 0.21 | 0.79 | 0.001 |
| Positive Affect | Within-subjects | 2.55 | 0.08 | 0.007 |
| | Between Subjects | 2.26 | 0.11 | 0.007 |
| | Interaction | 2.57 | 0.08 | 0.007 |
| **Effect of noticing trees** | | | | |
| Negative Affect | Within-subjects | 45.43 | < 0.01 | 0.11 |
| | Between Subjects | 3.85 | 0.05 | 0.01 |
| | Interaction | 0.15 | 0.7 | < 0.005 |
| Positive Affect | Within-subjects | 2.4 | 0.01 | 0.006 |
| | Between Subjects | 1.56 | 0.21 | 0.004 |
| | Interaction | 4.38 | 0.01 | 0.01 |
| **Effect of the time spent outdoor as a child** | | | | |
| Negative Affect | Within-subjects | 93.61 | < 0.005 | 0.2 |
| | Between Subjects | 1.42 | 0.24 | 0.008 |
| | Interaction | 0.37 | 0.81 | 0.2 |
| Positive Affect | Within-subjects | 3.39 | 0.03 | 0.009 |
| | Between Subjects | 1.74 | 0.18 | 0.009 |
| | Interaction | 0.36 | 0.83 | 0.002 |
| **Effect of Nature Connectedness** | | | | |
| Negative Affect | Within-subjects | 74.89 | < 0.005 | 0.17 |
| | Between Subjects | 0.49 | 0.81 | 0.008 |
| | Interaction | 0.65 | 0.69 | 0.01 |
| Positive Affect | Within-subjects | 6.18 | < 0.005 | 0.016 |
| | Between Subjects | 2.95 | 0.008 | 0.05 |
| | Interaction | 1.46 | 0.12 | 0.02 |

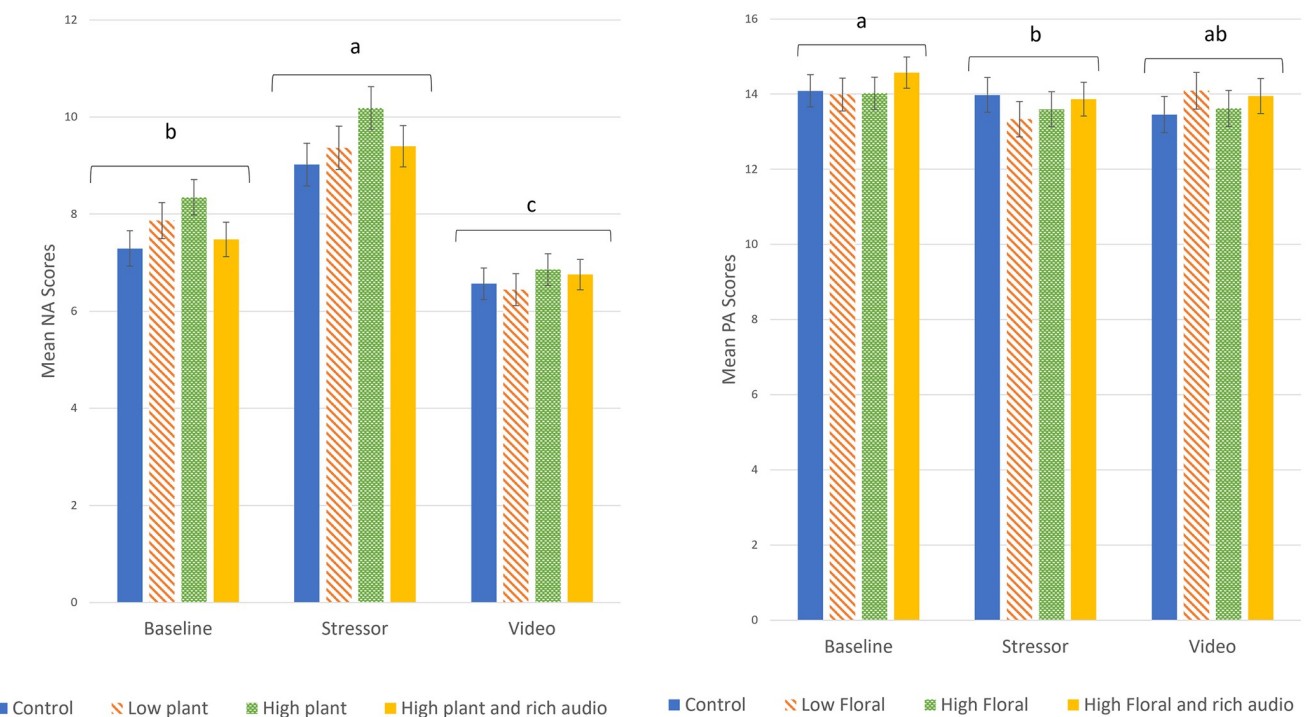

**Fig 3. Mean Negative Affect (Left) and Positive Affect (Right) scores by condition group at the three stages of the procedure.** Means without common letters are statistically different at p<0.05. Error bars represent +/- 1 standard error.

### Perceived level of biodiversity

*Perceptions* of the biodiversity shown in the videos showed an overall significant effect on both Negative and Positive Affect scores (NA p = 0.03; PA p < 0.005, Table 3). Post hoc tests showed that those who deemed the environment as "very good" for biodiversity had overall lower NA scores than those who rated it "neutral or bad" (p = 0.048) and overall higher PA scores than those who rated the environment "neutral or bad" (p = 0.008) or "good" (p = 0.001). The greatest reductions in NA between the post-stress and post-video stages were associated with those respondents who thought the sites were 'good' or 'very good' in terms of biodiversity (Fig 4, Left). Conversely, PA scores continued to decrease after the video for those who considered the environment "bad or neutral", but PA increased in those who thought the environment was "very good" in biodiversity (Fig 4, Right).

### Appreciating individual elements of nature

A large number of participants (208 individuals—59%) commented on their perceptions of biodiversity and natural features after watching the videos. Sounds were mentioned most frequently (N = 55, including wind, water and birds), then flowers (N = 35) and trees (N = 33) (S1 Appendix).

There was little overall difference in both Affect scores between those who mentioned noticing the flowers and the rest of the participants (NA p = 0.79; PA p = 0.19, Table 3). However, there was a significant interaction (p = 0.005) for Positive Affect. Those individuals who specifically commented about the flowers reported enhanced Positive Affect scores after watching the videos (Fig 5). Since all those who provided comments also perceived the

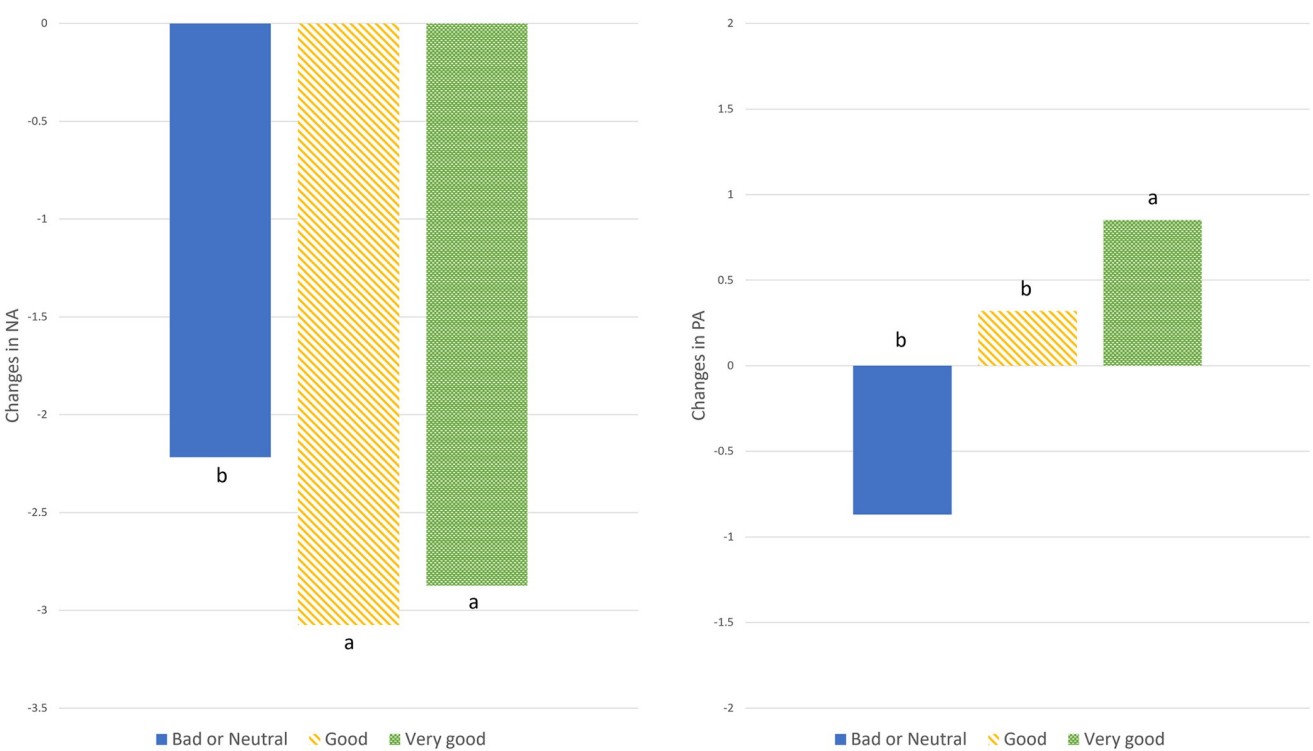

**Fig 4. Changes in Negative Affect (Left) and Positive Affect (Right) scores after the video stage for those groups who perceived the biodiversity as 'Bad/Neutral', 'Good' or 'Very Good'.** Letters denote statistical differences.

biodiversity to be "good" or "very good", those who noticed the flowers were compared with those who did not (i.e. controlling for the effect of perceived biodiversity). The interaction was again significant (p = 0.045), confirming that those who noticed the flowers showed significantly higher Positive Affect compared with those who noticed other things.

Participants who commented on natural sounds, including both the sound of the breeze and birdsong, reported significantly lower NA scores (p = 0.01) than the rest of the participants. The overall difference in PA scores, on the other hand, was non-significant (p = 0.11).

Participants who noticed the trees reported both lower Negative Affect (p = 0.05, Table 3) and lower Positive Affect scores overall (p = 0.21), compared to the other participants. The analysis of Positive Affect highlighted a significant interaction (p = 0.01). Participants who noticed the trees showed significantly higher scores in Positive Affect after the video, compared to the rest of the participants. However, when controlling for the effect of perceived biodiversity (see above) the interaction was no longer significant (p = 0.09).

## Time spent outdoors and nature connectedness

The time spent outdoors during childhood did not have a significant effect on the scores for Negative or Positive Affect (NA p = 0.24; PA p = 0.18, Table 3).

The level of Nature Connectedness had no influence on NA scores (p = 0.81), but showed a significant overall effect on PA scores (p = 0.008). Those moderately connected to nature had significantly higher PA scores than those who considered themselves poorly connected (INS = 5 vs INS = 1; p = 0.049). The least nature-connected participants reported decreasing PA scores throughout the procedure.

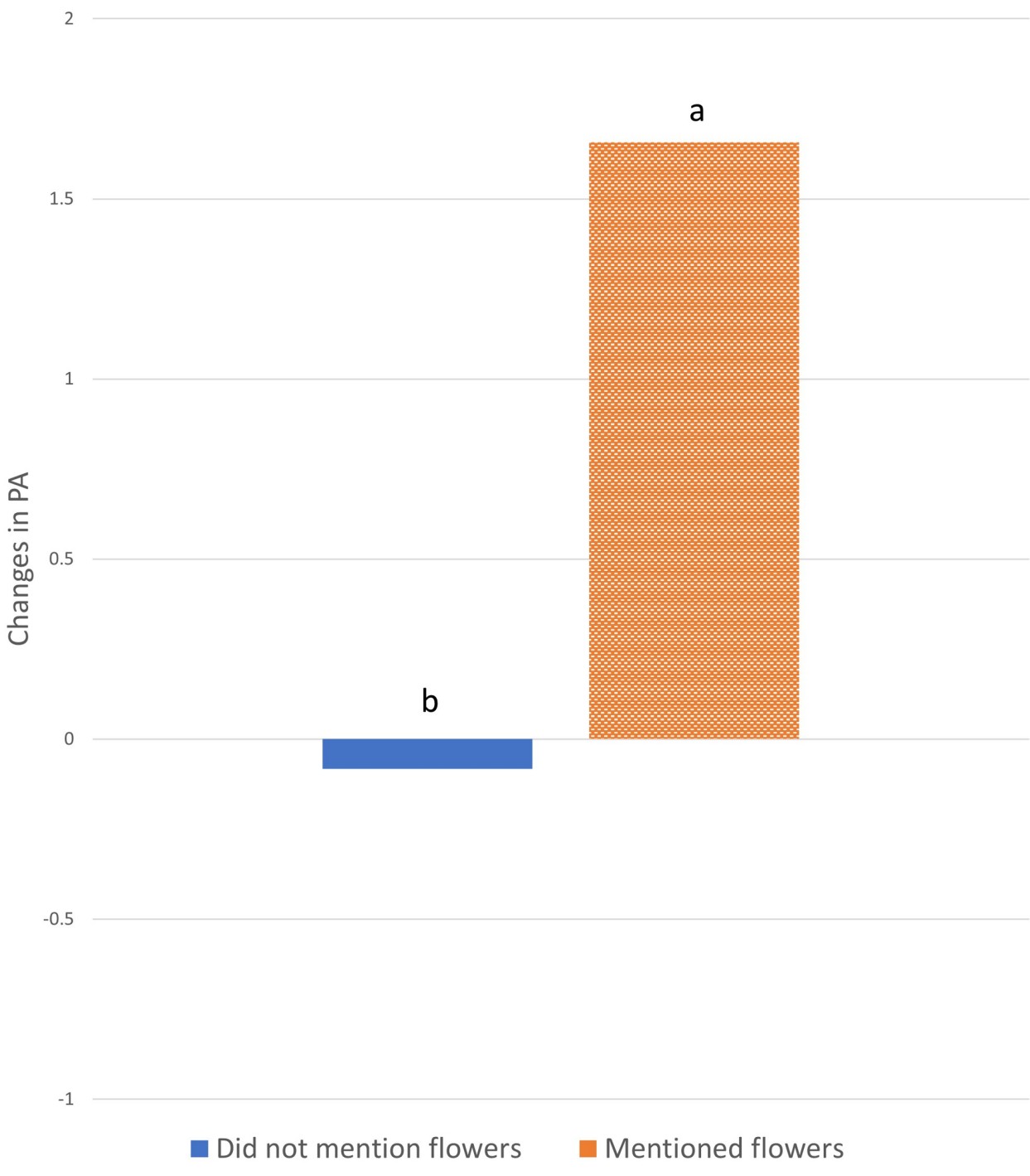

**Fig 5. Changes in mean positive affect scores after the video stage by those who mentioned the flowers and those who did not.** Letters denote statistical differences.

## Discussion

### Emotional responses to green space via video

This research supports the notion that exposure to green space (a video of urban woodland space) reduces feelings of negative emotion (Negative Affect), but does not necessarily increase

positive emotions (uplift–Positive Affect) *per se*. Scores for Negative Affect were lower after the video than at the start of the procedure or immediately after an annoying noise (stressor). This confirms previous findings that viewing green space reduces psychological stress [4, 5], including situations where the exposure is through video alone [47, 67, 68]. Previous studies with video have indicated no difference based on the type of green space being viewed [44], but Tyrväinen et al. [69] working with real *in vivo* urban spaces, suggested that typology mattered (urban woodlands being marginally more restorative than urban parks, with both typologies being better than a city centre location). It is possible that real locations stimulate stronger emotional responses than virtual ones [51], but real locations expose people to additional variables, other than those solely linked to the green infrastructure–not least different experiences whilst travelling to experimental locations. This study attempted to limit such variables, by focussing on one location and systematically altering the natural elements within. Whilst all our 'green' scenarios were equally restorative (lowering Negative Affect), none actually strongly stimulated joy (Positive Affect scores being lower after both the stressor and the video, compared to the baseline). This suggests our landscapes were providing a calming influence, but not necessarily stimulating strong feelings of excitement or joy.

## Biodiversity levels and emotions

Increasing the levels of plant biodiversity and inferring greater avian biodiversity through birdsong, had no additional significant effect on the affective response. In essence, there was no additional benefit from increasing these aspects of biodiversity to the green space. These results do not support our initial hypotheses that increased biodiversity improves the emotional (and potentially health) outcomes. Despite artificially altering distinct visual levels in plant (+21 species vs +4 species vs base) and sound levels of avian diversity (6 species vs none), we observed similar changes in the Affect scores as observed with the control treatment. Thus, for the sample population as a whole, the data does not support the notion that increased plant diversity corresponds to enhanced Positive Affect, and reduced Negative Affect, after a stressful event. Similarly, introducing background natural sounds (bird song and water) did not enhance Positive, or reduce Negative, Affect. Our findings are in line with those previous studies where either an increase in biodiversity [32, 70]–or optimal biodiversity [71]–did not increase the health outcomes.

This contrasts with the relationships between Positive Affect and biodiversity that have been found in other studies though. For instance, Wolf *et al.* [18] exposed their participants to videos with high vs low tree diversity and high vs low bird diversity. In both cases, they found a statistically significant effect for biodiversity and a higher post-video Positive Affect was associated with the high biodiversity conditions. Cameron *et al.* [19] also found significant positive correlations for both avian diversity and habitat diversity and the positive emotions reported by park visitors in Sheffield.

The absence of a direct biodiversity effect in this study could be due to non-exposure to the condition, rather than a non-response. It could be argued that the introduced species were not noticeable enough. However, the addition of brightly coloured, flowering plants was far from subtle. Comparing the Control condition with the High plant increase condition (Fig 1) it is evident that the plants stand out from the background. The same can be said for the avian diversity, where a rich soundscape was compared with the sound of the wind. It is possible, however, that the participants may not have paid much attention to the planting around them, focusing instead on the rest of the woodland. This would explain why, despite their actual differences in species, all videos received similar ratings of perceived biodiversity (S1 Appendix). This "nature myopia" has already been noted in previous experiments with flower meadows,

where the participants hardly noticed the difference in species, compared with control sites; nonetheless, those exposed to highly diverse meadows expressed a stronger preference [40] and higher site satisfaction [30] compared to controls.

## Perceived biodiversity and affective response

In contrast to data on real biodiversity, the *perceived* level of biodiversity was associated with increased Positive Affect scores. Those who perceived the environment as more biodiverse showed an increase in their Positive Affect after the video. These results align with previous research which hypothesised a predominant role for subjective perceptions of biodiversity over the actual species richness of the location [23, 32, 33]. In other words, what was visually *perceived* to be more diverse was more strongly associated with the Positive Affect scores than the *actual* diversity.

However, only half of the commenters (108 out of 208) associated their high perceptions of biodiversity with elements related to species richness (i.e. trees, birds, flowers) when asked. Fewer could narrow down their perceptions to birds (18 comments) or flowers (35 comments) which were experimentally designed to be evident. Those participants who noticed flowers, trees or sounds showed improved Affect scores after watching the video (i.e. reduced Negative Affect and increased Positive Affect). These participants received something more from the experience by *engaging* with the local species richness (i.e. trees) and the species we had introduced (i.e. flowers and birdsongs). This suggests that noticing biodiversity could be a necessary step to receive extra emotional recovery after a stressful event.

Research shows that actively noticing nature is more beneficial than passive exposure. Randomised controlled trials such as the Noticing Nature Intervention [72] and its replications [73, 74] showed that participants who were tasked to notice nature for two weeks reported higher Positive Affect, compared with participants who did not receive instructions. This increase in Positive Affect was independent of other well-being-related variables, such as Nature Connectedness and engagement with beauty. Similarly, noticing nature with all senses during a forest bathing session [75] improved Positive Affect as much as a session of Compassionate Mind Training, which is a more established intervention to improve well-being.

Among the natural elements that participants used to evaluate the video's biodiversity, *flowers* deserve a special mention. Flowers were the only biodiversity element correctly noticed by participants (i.e. there were no comments about flowers in the control condition). Noticing flowers was associated with improved Positive Affect, even when the effect of the perceived biodiversity was statistically controlled. Flowering plants are known to elicit positive emotions [76–78], especially when the flower coverage exceeds a certain percentage (e.g. 27% [79]). Although in this case it is difficult to estimate the scene coverage, due to its three-dimensional nature, if the participants spent most of the condition time looking at the planted flowers, this could have prompted the increase in Positive Affect.

## Limitations and recommendations for future research

As the study was conducted online, there are limitations in estimating how much the participants engaged with the experimental procedure. Although metadata provided a measure of control on some of the factors (e.g. individuality of the response, type of device used), it is challenging to say whether the participants followed the recommendations provided. Using a 360-degree video increased the simulation of "being there" but at the cost of some control over the condition. Since the participants could point the camera in any direction, some could have chosen to look at the tree canopy for the entire video, without noticing the ground flowers. Furthermore, participants did not receive any instructions other than to "look around". It is

possible that the differences in Positive Affect between commenters and non-commenters (who represented over 40% of the sample) might be explained by boredom with the lengthy procedure. However, even though boredom was mentioned in the comments by 30 participants, the completion rate suggests that most of the participants did their best to engage with the experiment.

There were notable differences in results from people who noticed elements of nature and those who did not (or did not report doing so). Future research should focus on this aspect. How much do we need to notice nature–to gain a restorative benefit? Are there differences due to more nuanced demographics? Future research could take a stratified approach to determine if nature engagement/knowledge impacts on well-being. Asking people to notice nature seems to help with engagement and subsequently well-being [72–74], but how do background levels of nature literacy impact on emotional responses? It may well be that both a complete lack of knowledge/interest (not noticing or valuing what is seen) or high levels of nature knowledge (e.g. in conservation workers with consequential understanding of negative factors–biodiversity declines, habitat loss etc.) act negatively in terms of promoting well-being. Perhaps only those with a moderate understanding of nature actually have the capacity to receive the health benefits? These factors need testing.

## Conclusions

This study explored the impact of biodiversity on emotional responses but differed from many previous studies by artificially manipulating, in a controlled manner, the amount of biodiversity on view or inferred through sound. Although the study noted that being in a green (woodland) space decreased negative emotions, there was no significant additional benefit due to increased biodiversity. Being exposed to green space (and different levels of biodiversity) had little impact on the Positive Affect participants reported, but *perceiving* the location as more biodiverse had a greater effect. Those participants who had positive perceptions about the environment (e.g. how biodiverse they thought it was) and their engagement with it (by noticing and appreciating its elements) reported more Positive Affect. Noticing biodiversity, therefore, could be a key factor in eliciting positive emotions, however, fewer than 10% of participants indicated they noticed any additional plantings (flowering plants). If perceiving and engaging with biodiversity is necessary to receive an extra boost to mental well-being, a lack of noticing may be preventing many people from gaining the maximum mental health benefit from their green space. In essence, whilst acknowledging some psychological recovery provided by green space in general, one may have to "notice" the good things in nature to optimise those benefits. This data has important implications for policymakers, because not only do urban green spaces need to be engaging and biodiverse, but certain educational processes may need to be included before citizens reap the health benefits such places can elicit. In essence, "nature blindness" may undermine the salutogenic potential of many green spaces.

## Supporting information

**S1 Appendix. Supporting information.** Species lists, codes description and perceived biodiversity crosstabulation.
(DOCX)

**S1 Dataset. Dataset and statistics outputs.**
(ZIP)

## Acknowledgments

The authors would like to thank Ms Liwen Zhang and Mr John Land for their comments on the manuscript.

## Author Contributions

**Conceptualization:** Simone Farris, Nicola Dempsey, Kirsten McEwan, Helen Hoyle, Ross Cameron.

**Data curation:** Simone Farris.

**Formal analysis:** Simone Farris, Nicola Dempsey, Ross Cameron.

**Investigation:** Simone Farris, Ross Cameron.

**Methodology:** Simone Farris, Nicola Dempsey, Kirsten McEwan, Ross Cameron.

**Supervision:** Nicola Dempsey, Kirsten McEwan, Ross Cameron.

**Writing – original draft:** Simone Farris, Ross Cameron.

**Writing – review & editing:** Simone Farris, Nicola Dempsey, Kirsten McEwan, Helen Hoyle, Ross Cameron.

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
