## [Decision Letter · Decision Letter 0]

24 Aug 2023

PONE-D-23-09550Does increasing biodiversity in an urban woodland setting, promote positive emotional responses in humans? A Stress Recovery Experiment Using 360-degree Videos of an Urban WoodlandPLOS ONE

Dear Dr. Farris,

Thank you for submitting your manuscript to PLOS ONE. After careful consideration, we feel that it has merit but does not fully meet PLOS ONE’s publication criteria as it currently stands. Therefore, we invite you to submit a revised version of the manuscript that addresses the points raised during the review process.

ACADEMIC EDITOR: The comments are added at the end. 

We look forward to receiving your revised manuscript.

Kind regards,

Bidisha Banerjee, Ph.D.

Academic Editor

PLOS ONE

Journal Requirements:

2. In your Methods section, please provide additional information regarding the permits you obtained for the work. Please ensure you have included the full name of the authority that approved the field site access and, if no permits were required, a brief statement explaining why

Reviewers' comments:

Reviewer's Responses to Questions

**Comments to the Author**

1. Is the manuscript technically sound, and do the data support the conclusions?

Reviewer #1: Yes

Reviewer #2: Yes

2. Has the statistical analysis been performed appropriately and rigorously? 

Reviewer #1: Yes

Reviewer #2: Yes

3. Have the authors made all data underlying the findings in their manuscript fully available?

Reviewer #1: No

Reviewer #2: Yes

4. Is the manuscript presented in an intelligible fashion and written in standard English?

Reviewer #1: Yes

Reviewer #2: Yes

5. Review Comments to the Author

Reviewer #1: This work depicts the impact of biodiversity on stress reduction and enhancing positive emotional well-being. The impact is being assessed by the actual richness of biodiversity and the perception of the richness of biodiversity. Despite its merits, I recommend addressing the following aspects in a revised paper version.

Abstract

The participants have recorded perceptions of biodiversity, but it is not listed as one of the study's objectives.

Line 29: Change "their on" to "on their" (There are many such instances, please read and make changes)

Introduction

The introductory paragraph may also include about perception (with an explanation) of biodiversity, which is addressed in the later parts.

Paragraphs 4 and 5 (lines 58 to 69): Consists of all the findings from the literature. Presenting these findings separately with an explanation is suggested.

Are there any similar studies conducted using similar methods? If yes, pls cite them.

The description of the study conducted can be organized under the sub-heading "the present study" or similar.

Method

Overall, it looks unorganized.

Line 222 to 225: Can be part of "procedure"

Line 2248 to 251: Include under the details of the measures used

Results

Presenting the statistical results in a table would be easier for readers to understand.

Discussion can be more rigorous, specific to the Results of the study.

Recommendations for future research- Not Mentioned

Others- Table formats are not uniform

Reviewer #2: The manuscript was reviewed for the incorporation of suggested comments made by the previous reviewers. After reviewing the comments and authors' reply, it has been found that authors have incorporated all comments in the manuscript. I do not see any concerns related to research ethics and publication ethics.

6. PLOS authors have the option to publish the peer review history of their article (what does this mean?). If published, this will include your full peer review and any attached files.

Reviewer #1: No

Reviewer #2: No

---

## [Author Response · Author response to Decision Letter 0]

22 Sep 2023

Response to the editor’s comments

(Note. Line numbers mentioned in this response refer to the unmarked version of the revised manuscript without tracked changes)

Comment 1). Please ensure that your manuscript meets PLOS ONE's style requirements, including those for file naming.

The manuscript has now been formatted according to the journal’s style.

Comment 2). In your Methods section, please provide additional information regarding the permits you obtained for the work.

A statement concerning the permission obtained for the study has been added (Lines 150-151)

Commetn 3). Please include captions for your Supporting Information files at the end of your manuscript, and update any in-text citations to match accordingly.

Captions have been added, as requested. (Lines 512-514)

Comment 4). Please review your reference list to ensure that it is complete and correct.

The reference list has been reviewed. The style is now uniform and in line with the journal’s specifications.

All figures’ files have been processed with PACE

Response to the reviewers’ comments

Comment 1). Have the authors made all data underlying the findings in their manuscript fully available? Reviewer #1: NO Reviewer #2: YES

A dataset file is publicly available from the UK Data Service database

URL: https://reshare.ukdataservice.ac.uk/856311/

DOI: https://dx.doi.org/10.5255/UKDA-SN-856311

This dataset contains demographic information, computed values of the dependent variables, ratings of nature connectedness and perceived biodiversity, original textual responses from the participants and dummy variables derived by the content analysis. The dataset file is also available as part of the supporting information attached to the submission (Dataset and statistics output.zip), along with the statistical analysis outputs. 

Abstract

Comment 2). The participants have recorded perceptions of biodiversity, but it is not listed as one of the study's objectives.

We thank reviewer #1 for this suggestion. The abstract now mentions perceived biodiversity as part of the aims (line 21). Perceptions of biodiversity were not part of the original objectives of the study, but a measure of perceptions was included to act as a control, informed by the studies cited in lines 93-99.

Comment 3). Line 29: Change "their on" to "on their" (There are many such instances, please read and make changes)

Changed. The text is now thoroughly proofread and all these typos have been corrected.

 

Introduction

Comment 4). The introductory paragraph may also include about perception (with an explanation) of biodiversity, which is addressed in the later parts.

Perceived biodiversity, and a definition of it, is now mentioned in the second paragraph after the concept of biodiversity is introduced (lines 51-54). It is more logical to place it here, as the first paragraph deals with the health benefit theories alone.

Comment 5). Paragraphs 4 and 5 (lines 58 to 69): Consists of all the findings from the literature. Presenting these findings separately with an explanation is suggested. Are there any similar studies conducted using similar methods? If yes, pls cite them.

We thank reviewer #1 for these suggestions. New summaries of previous experiments involving biodiversity have been added in Paragraphs 4 and 5. All the six studies mentioned in line 66 are now explained, making the review more systematic. Several of these studies have been mentioned for their use of similar methods. E.g. references 18 and 31 used 360-degree videos and references 29-31 have repeated measure designs. We intended to both highlight the paucity of experimental evidence and some limitations of previous studies, in particular the comparison of different locations.

Comment 7) The description of the study conducted can be organized under the sub-heading "the present study" or similar.

A subheading has been introduced (line 118)

Method

Comment 8). Line 222 to 225: Can be part of "procedure"

We thank reviewer #1 for this suggestion. However, we would argue that the description of the procedure already mentions the collection of demographic information (line 194). In the context of the “measures” section, lines 237-240 explain what demographic information was collected and why.

Comment 9). Line 248 to 251: Include under the details of the measures used

This paragraph has been moved to the “measures” section, lines 225-228.

Results

Comment 10). Presenting the statistical results in a table would be easier for readers to understand.

The results of the repeated measures ANOVAs have been summarised in a table (Table 3, line 307) to allow full comprehension of the statistics, whilst retaining the pertinent ‘p’ significant values in the text.

Comment 11). Discussion can be more rigorous, specific to the Results of the study.

The discussion has been rearranged to better discuss the three main findings: 1) Emotional responses to green space via video; 2) biodiversity levels and emotions; 3) Perceived biodiversity and affective response. 

Comment 12). Recommendations for future research- Not Mentioned

We thank reviewer #1 for highlighting this. We introduced a few suggestions for future research. Lines 469-480

Comment 13). Others- Table formats are not uniform

All tables are now in the same format, in line with the journal’s guidelines

---

## [Decision Letter · Decision Letter 1]

2 Jan 2024

Does increasing biodiversity in an urban woodland setting, promote positive emotional responses in humans? A Stress Recovery Experiment Using 360-degree Videos of an Urban Woodland

PONE-D-23-09550R1

Dear Dr. Farris

We’re pleased to inform you that your manuscript has been judged scientifically suitable for publication and will be formally accepted for publication once it meets all outstanding technical requirements.

If your institution or institutions have a press office, please notify them about your upcoming paper to help maximize its impact. If they’ll be preparing press materials, please inform our press team as soon as possible -- no later than 48 hours after receiving the formal acceptance. Your manuscript will remain under a strict press embargo until 2 p.m. Eastern Time on the date of publication. For more information, please contact onepress@plos.org.

Kind regards,

Tunira Bhadauria, Ph.D.

Academic Editor

PLOS ONE

Additional Editor Comments (optional):

Reviewers' comments:

Reviewer's Responses to Questions

**Comments to the Author**

1. If the authors have adequately addressed your comments raised in a previous round of review and you feel that this manuscript is now acceptable for publication, you may indicate that here to bypass the “Comments to the Author” section, enter your conflict of interest statement in the “Confidential to Editor” section, and submit your "Accept" recommendation.

Reviewer #2: All comments have been addressed

Reviewer #3: All comments have been addressed

2. Is the manuscript technically sound, and do the data support the conclusions?

Reviewer #2: Yes

Reviewer #3: Yes

3. Has the statistical analysis been performed appropriately and rigorously? 

Reviewer #2: Yes

Reviewer #3: Yes

4. Have the authors made all data underlying the findings in their manuscript fully available?

Reviewer #2: Yes

Reviewer #3: Yes

5. Is the manuscript presented in an intelligible fashion and written in standard English?

Reviewer #2: Yes

Reviewer #3: Yes

6. Review Comments to the Author

Reviewer #2: Dear Author(s). Thank you for addressing our concerns and answering our queries. Your study seems interesting. All the best.

Reviewer #3: I reviewed the manuscript entitled "Does increasing biodiversity in an urban woodland setting, promote positive emotional responses in humans? A Stress Recovery Experiment Using 360-degree Videos of an Urban Woodland".

I would like to extend my congratulations to the authors for their comprehensive study. In general, the manuscript is well-crafted, presenting information in an easily digestible manner and employing sophisticated analyses. The findings are notably intriguing. Nevertheless, I have some suggestions that may enhance the clarity and comprehensibility of the text.

Lines 204 - 205: It would be beneficial to provide a brief explanation of what the I-PANAS-SF entails.

Line 209: It may be informative to specify the number or percentage of non-native English speakers who participated in the study.

Lines 239 – 247: Including information about how the data adheres to the assumptions of One-way analysis of variance (ANOVA) and Repeated measure analysis of variance (RM-ANOVA) would improve the robustness of your statistical analysis.

Lines 246 - 247: In the statement, "Where post-hoc comparisons were performed, the Bonferroni correction for multiple comparisons was applied," it would be helpful to include a reference regarding the Bonferroni correction for the benefit of readers.

Lines 250 - 251: When stating, "The internal reliability, measured as Cronbach alpha, was good for both scales (Cronbach α > 0.8)," it would be advantageous to provide a reference that supports this assessment.

Furthermore, it would be intriguing if the authors could explore potential variations in emotional responses among participants with different demographic characteristics, such as gender, age, and ethnicity, as this could provide valuable insights into the study's outcomes.

7. PLOS authors have the option to publish the peer review history of their article (what does this mean?). If published, this will include your full peer review and any attached files.

Reviewer #2: No

Reviewer #3: No

---

## [Editor Report · Acceptance letter]

12 Jan 2024

PONE-D-23-09550R1 

PLOS ONE

Dear Dr. Farris, 

I'm pleased to inform you that your manuscript has been deemed suitable for publication in PLOS ONE. Congratulations! Your manuscript is now being handed over to our production team.

Kind regards, 

on behalf of

Dr. Tunira Bhadauria 

Academic Editor

PLOS ONE